# The Mechanism of Action between Pulsed Radiofrequency and Orthobiologics: Is There a Synergistic Effect?

**DOI:** 10.3390/ijms231911726

**Published:** 2022-10-03

**Authors:** Daniel de Moraes Ferreira Jorge, Stephany Cares Huber, Bruno Lima Rodrigues, Lucas Furtado Da Fonseca, Gabriel Ohana Marques Azzini, Carlos Amilcar Parada, Christian Paulus-Romero, José Fábio Santos Duarte Lana

**Affiliations:** 1Instituto Regenem de Medicina, Rua Bandeira Paulista 716, São Paulo 04532-0021, Brazil; 2Orthoregen International Course—Avenida Presidente Kennedy, 1386, Cidade Nova I, Indaiatuba 13334-170, Brazil; 3Orthopaedic Department, Universidade Federal de São Paulo, 715 Napoleão de Barros St-Vila Clementino, São Paulo 04024-002, Brazil; 4Department of Orthopaedics, Brazilian Institute of Regenerative Medicine, Cidade Nova I, Indaiatuba 13334-170, Brazil; 5Laboratory of Study of Pain, Department of Structural and Functional Biology, University of Campinas, Rua Monteiro Lobato, 255, Campinas 13083-862, Brazil; 6American Academy of Regenerative Medicine, 14405 West Colfax Avenue, #291, Lakewood, CO 80401, USA

**Keywords:** pulsed radiofrequency, orthobiologics, neuromodulation, growth factors

## Abstract

Radiofrequency energy is a common treatment modality for chronic pain. While there are different forms of radiofrequency-based therapeutics, the common concept is the generation of an electromagnetic field in the applied area, that can result in neuromodulation (pulsed radiofrequency—PRF) or ablation. Our specific focus relates to PRF due to the possibility of modulation that is in accordance with the mechanisms of action of orthobiologics. The proposed mechanism of action of PRF pertaining to pain relief relies on a decrease in pro-inflammatory cytokines, an increase in cytosolic calcium concentration, a general effect on the immune system, and a reduction in the formation of free radical molecules. The primary known properties of orthobiologics constitute the release of growth factors, a stimulus for endogenous repair, analgesia, and improvement of the function of the injured area. In this review, we described the mechanism of action of both treatments and pertinent scientific references to the use of the combination of PRF and orthobiologics. Our hypothesis is a synergic effect with the combination of both techniques which could benefit patients and improve the life quality.

## 1. Introduction

Radiofrequency (RF) energy-based procedures, whether conventional, ablative or pulsed, represent a technique commonly performed for chronic pain in a variety of musculoskeletal conditions [1,2,3].

Pulsed radiofrequency (PRF) is derived from conventional RF with the aim of a less destructive RF-based treatment to be applied to the afferent nerve pathways of injured tissues [4]. PRF creates an electromagnetic field with the aim of functionally disrupting the neuronal membrane, which modulates gene expression, affecting the release of cytokines [5]. The application of PRF is based on the delivery of a train of sinusoidal electrical bursts (5–20 ms length) in the radiofrequency range (500 kHz) at a repetitive rate of a few hertz (2–5 Hz) [6] (Figure 1).

The changes produced by electrical fields are selective for small unmyelinated and lightly myelinated nerve fibers, producing a motor-sparing effect [5]. Histological evaluations show that PRF promotes transient endoneural edema that can persist for up to 1 week following treatment [7]. The pain relief commonly seen after PRF treatment can last up to several months [8]. Currently, the most common pathologies treated with PRF are radicular pain, occipital and trigeminal neuralgia, and shoulder and knee pain [9].

Although both techniques have been reported to be effective (pulsed and ablative RF), there are some limitations more related to the ablative RF, such as failure to completely denervate the nerve, disrupt nociceptive sensation, worsening of symptoms due to aberrant neuronal regeneration, neuroma formation, and reduced motor function, neuritis, paresthesia and deafferentation pain syndrome [10].

## 2. Pulsed Radiofrequency: Mechanisms of Action

Some studies demonstrated that the analgesic effect of PRF is not related to thermal effects or to permanent physical neural damage [11,12]. This effect could be due to a neuromodulatory-type process, which alters the synaptic transmission or the excitability of C-fibers [5,13]. These fibers are responsible for pain and temperature sensations and are involved in most neuropathic pain syndromes [14].

Preclinical studies have proposed several biological effects of PRF. The mechanisms of action may consist of morphological changes in the inner structures of axons [15], molecular effects, including alterations in cellular activity [16], gene expression [8,17,18], an increase in the expression of inflammatory cytokines [15] and inhibition of extracellular signal-regulated kinases [19]. Recently, it was demonstrated that PRF may have a long-term depression effect on neuropathic pain [20]. These reported mechanisms help to elucidate how PRF inhibits the transmission of pain signals in a biological pathway. However, the biophysical mechanisms by which this electric field may act are not clearly elucidated yet [6]. One of the main effects reported after a PRF treatment is the increase in cytosolic free calcium concentration, an important messenger involved in both short and long-term cellular processes. This mechanism could link PRF effects to a direct consequence on electric fields [6].

In addition, it was hypothesized that PRF could cause neural membrane permeabilization due to a mild electroporation process leading to a Ca^2+^ influx. Once the cell is exposed to high electric fields for a short period, the cell membrane increases its permeability to ions and molecules, a process called electroporation. Accordingly, PRF could be applied to different regions of afferent nervous pathways, so this influx could occur in the same fashion in all aspects of the targeted neuron [21].

PRF is not limited to targeting afferent nerves, which was observed after the use of intra-articular PRF, and, as a result, it was verified that PRF promotes an analgesic effect. Therefore, it was suggested that PRF could present a local anti-inflammatory effect due to an effect on the immune system, which may ultimately impact the nociceptive process [22,23].

It was reported that PRF has the ability to decrease levels of interleukin-1 (IL-1), metalloproteinase-3 (MMP-3), and tumor necrosis (TNF-α) in the synovial fluid of severe osteoarthritic (OA) patients. This was demonstrated by the observation of clinical improvements, resulting in better outcomes in comparison to the use of betamethasone. The effect on the immune response could be explained by the inhibition of immune cells and pro-inflammatory cytokines. Therefore, as inflammatory cytokines are regulated, PRF stimulates a greater level of cascade reaction-stop amplification of the inflammatory reaction and avoids the common inflammatory wind-up phenomenon. This may explain the mechanisms of long-term pain mitigation after using PRF [24]. Given its ability to impact immune cells, there are case reports considering the intravenous route to apply RF in order to treat unresolved immune issues [22].

Preclinical studies demonstrated a rapid onset (within 3 h) increase in c-FOS expression, a specific marker for cellular activity, which lasted one week after PRF stimulus. c-FOS inhibition of excitatory C fibers is a possible mechanism involved in analgesia [25]. In vitro, PRF stimuli induce a transient decrease in excitatory postsynaptic potential, with recovery in a fast and complete way in hippocampal organotypic slices [26].

Pertaining to ultrastructural changes in the sciatic nerve, a preclinical study reported extensive mitochondrial swelling and hyperplasia. Functional recovery of the animals suggests that this mechanism is a compensatory response that helps the recovery and regeneration of lesioned nerve fibers. In the treated animals, it was observed that the macrophages contained intracellular cholesterol crystals and necrotic tissue, suggesting that PRF promoted an inflammatory response to clear the lesion area, as an early response to PRF [17].

Up until now, the literature has been trying to elucidate the mechanisms of action underlying the use of PRF. Well-designed randomized controlled and in vitro trials are needed in order to clarify the mechanism of action further for procedures at frequencies and temperatures used in current clinical practice and how they modify central and peripheral components of pain pathways. Additionally, it is important to note that clinical outcomes could be influenced by lesion parameters (sensory and motor stimulation thresholds), lesion duration, electrode position, and local tissue properties [27].

Another possible effect of PRF involves changes in the oxygen molecule. The recombination of oxygen (geminate recombination or cage recombination) competes with the final separation of the radicals (escape reaction with the possibility of forming products different from those of cage recombination). The ratio of the cage to escape reaction yields will critically depend on the rate of spin evolution, which, on the other hand, depends on an external magnetic field [28].

Thus, the reaction kinetics become magnetic field dependent and this could explain the phenomena that result in the PRF mechanism of action. The radical pair mechanism is a plausible way in which weak magnetic field variations can affect chemical reactivity, allowing radical pairs containing substances that can function as chemical/biological magnetic sensors [28,29].

Regarding the analgesic effects of PRF in nerves and other tissues, we summarized them in Table 1.

This review will focus on PRF and its potential synergistic action with orthobiologics, given that PRF represents a less destructive and more modulatory role, which is in agreement with the mechanisms of the orthobiological substances.

## 3. Orthobiologics

Orthobiologics comprise a group of autologous biological materials, including bone grafts, growth factors, cell-signaling proteins, and cell-based therapies, such as platelet rich plasma (PRP), bone marrow- and adipose tissue-derived products, which promote bone, ligament, muscle, and tendon healing [31]. The wide spectrum of efficacy make orthobiologics a promising alternative for treating a diversity of orthopedic conditions. Orthobiolical substances have been indicated for bone healing, given their osteoconduction, osteoinduction, and osteogenesis properties [31], for osteoarthritis, and tendinopathy, among others [32,33,34].

Although the concept of biologics is not quite new—platelet rich plasma has been used in oral maxillofacial surgery for decades [35], most advances in this field have been made in the last two decades. A large number of promising studies show that these products have great potential to alter the next chapter of the evolution of orthopedics [31].

### 3.1. Platelet-Rich Plasma

Platelet rich plasma (PRP) is an autologous product with a platelet concentration above baseline in a small volume of plasma. It is obtained following steps of centrifugation and cell selection [35]. Its efficacy is based on the impact of platelets on wound healing as they release local growth factors and recruit repair cells to induce tissue regeneration [36].

Upon activation, platelets release molecules that bind to the specific receptors on the cell membrane of the target cells, triggering a signaling cascade that activates acute inflammation, with tissue repair as an end product [37]. Among these growth factors, there are platelet-derived growth factor (PDGF), which is responsible for stem cell stimulation, and fibroblast and immune cell recruitment; transforming growth factor β1 (TGF-β1), which promotes extracellular matrix synthesis; vascular endothelial growth factor (VEGF), which promotes angiogenesis and stimulates endothelial cell proliferation; endothelial growth factor (EGF), responsible for promoting epithelization, cell differentiation and maturation, tissue remodeling by influencing collagenase activity [37]. In addition, there is stromal-derived factor 1 (SDF-1), which assists the recruitment of stem cells [38].

PRP has been studied to treat a range of musculoskeletal conditions. A recent meta-analysis with 18 studies was published comparing the use of PRP with hyaluronic acid (HA) for the treatment of knee OA. An improvement was noted in the PRP group for the function scores of specific joints. In addition, patients who received PRP also reported significant pain relief [39]. Another meta-analysis stated PRP is a safe option to treat rotator cuff tendinopathy with long-term efficacy, improving shoulder symptoms and function [40].

The use of PRP for pain management is also a matter of discussion in the literature. Studies that evaluated PRP for the treatment of knee OA reported a significant decrease in the visual analogic scale (VAS) for pain evaluation and WOMAC pain score [34,41]. A few critical review studies have also reported an improvement in pain relief with the use of PRP for low back pain [42,43].

### 3.2. Bone Marrow-Derived Products

Bone marrow is a rich source of stem and progenitor cells, including hematopoietic cells (HSC) and mesenchymal stem cells (MSC) [44]. HSCs are responsible for the formation of the hematopoietic microenvironment and provide circulating mature blood cells, including erythrocytes, leukocytes and platelets, whereas MSCs are able to differentiate into cartilage, bone, fat, muscle, meniscus and tendon, which provides a fundamental ability for the regeneration process [45]. There are two main bone marrow-derived products: bone marrow aspirate (BMA) and bone marrow aspirate concentrate (BMAC). Recently, a classification of these products was proposed by our group [44].

Bone marrow aspirate (BMA) is a rich source of MSCs and HSCs and is also considered a great reservoir of growth factors [46]. BMA can be harvested without the use of anticoagulants, leading to the formation of a clot that plays an important role in tissue repair. Its use has been discussed in the literature for the treatment of musculoskeletal disorders [47]. Clot formation occurs due to platelet activation and degranulation, which leads to the release of osteotrophic cytokines and growth factors at the injured site. It can also aid in the stability of the cell graft at the injured site [48]. Furthermore, the fibrinolytic activity can lead to the release of angiogenic factors, an essential process for tissue repair initiation [49]. There are only a few studies using BMA, and most of them are regarding nonunions [50,51,52,53]. One study evaluated a combination of BMA with prolotherapy to treat knee, hip, and ankle osteoarthritis with a significant improvement seen in twelve months of follow-up [54].

Bone marrow aspirate concentrated (BMAC) is a product with anti-inflammatory and immunomodulatory characteristics, capable of potentially augmenting tissue repair as it presents high levels of all cellular and molecular components present in BMA [55]. The efficacy observed with BMAC is based on the concentration of its molecular content, which is composed of anabolic and anti-inflammatory molecules, including PDGF, TGF-β, VEGF, and bone morphogenetic protein (BMP) −2 and −7. Hence, this product provides an optimized cellular physiological environment for the promotion of osteogenesis and angiogenesis [55].

The efficacy of BMAC has been reported for several musculoskeletal conditions. Centeno et al. studied the effects of BMAC on shoulders with rotator cuff injuries and shoulder osteoarthritis. An improvement in joint function and disability and a decrease in pain were reported with both outcomes reaching statistical significance [56]. With regard to cartilage defects, Gobbi et al. reported a complete coverage of lesions with hyaline-like cartilage in most of the patients treated with a hyaluronic acid-based scaffold embedded with BMAC [57]. Rodriguez-Fontan and colleagues evaluated the efficacy of intra-articular BMAC injections for the treatment of early knee and hip osteoarthritis. It was found to be safe and demonstrated satisfactory results in 63.2% of patients [58].

### 3.3. Adipose Tissue-Derived Products

Adipose tissue (AT) is predominantly composed of mature adipocytes containing lipids, and stromal vascular cells, such as preadipocytes, interstitial cells, endothelial cells, mesenchymal cells and pericytes. In addition, AT presents an extensive system of blood vessels, lymph nodes, and nerves, associated with an extracellular matrix (ECM) composed of collagen types I, III, IV, V, and VI, and other ECM proteins [59].

AT is also recognized as a vital endocrine organ with the ability to secrete adipokines and inflammatory cytokines. Adipokines are represented by adiponectin, leptin, resistin, visfatin; inflammatory cytokines include TNF-α, IL-6 and IL-1 [60]. With this in mind, in the context of Regenerative Medicine, especially focusing on the orthopedic area, the use of whole adipose tissue as a fresh graft may not result in beneficial results.

Vascular stromal fraction (SVF) is a heterogeneous mixture of cells, including mesenchymal stem cells (AD-MSC), which play important roles in the regenerative context [61]. Several translational studies have indicated that SVF is able to promote tissue regeneration through a combination of cell-mediated repair mechanisms and their paracrine effect [62,63,64]

In orthopedics, the use of SVF has been indicated for several conditions. Most of the studies for cartilage regeneration associated SVF with other orthobiologics, such as PRP or HA, and they reported promising results: pain relief, improvement of function, and cartilage regeneration seen in magnetic resonance image (MRI) of patients with OA [65,66,67]. In addition, SVF has also been shown to be effective for chondromalacia patellae [68] and meniscus tears [69].

## 4. Brain-Derived Neurotrophic Factor (BDNF) Could Be a Critical Molecule to Determine PRP-PRF-Associated Treatment

Data from the literature now consider Brain-derived Neurotrophic Factor (BDNF) a critical factor involved in the anti-hyperalgesic effect mediated by PRF applied in the Dorsal Root Ganglion (DRG) or Spinal Cord. BDNF is a protein of the neurotrophin family. Neurotrophins are a family of growth factors that include, in addition to BDNF, nerve growth factor (NGF), neurotrophin-3 (NT-3), and neurotrophin 4/5 (NT-4/5). They were initially described as trophic factors involved in neuronal development, survival, and function [70,71]. Developmentally, they are involved in synapse formation and neuroplasticity. This involvement was demonstrated in some chronic pain conditions of the nervous systems where the formation of a new synapse and neuroplasticity are maladaptive [72,73,74,75]. BDNF exerts its physiological functions through two different types of receptors: Tyrosine kinase B (TrkB) with higher affinity and pan-neurotrophin receptor p75 (p75NTR) with lower affinity [76,77]. Outside the nervous system, neurotrophin receptors, including TrKB and P75 are expressed in tissues, such as skeletal muscle, bone, immune system cells (basophils, eosinophils, lymphocytes; macrophages; mast cells, neutrophils), cartilage, and synovium [77,78,79,80,81,82,83]. In this wide variety of tissues, neurotrophins play a constitutive role in regulating cell homeostasis and tissue development [77,80,82]. They exert different roles in every tissue, however, in general, TrKB receptors are related to cell differentiation, cell-matrix synthesis, and tissue homeostasis [79,80,84]. In addition, the activation of the TrKB receptor in macrophages induces the polarization of M2 macrophages and the synthesis of anti-inflammatory cytokines IL-10 and IL-13 [85,86,87,88], and in dendritic cells, the synthesis and release of IL-4 [89]. On the other hand, P75 receptors are related to a pro-inflammatory microenvironment with the release of MMP-2 metalloproteinase enzymes and the synthesis of pro-inflammatory cytokines, such as IL-1β, TNF-α, IL-6, and IL-8 [81], also, expression of MMP-9 metalloproteinase, and phagocytosis [90].

In summary, data from the literature describe that BDNF outside the nervous system via TrKB receptor activation promotes anti-inflammatory and immunomodulatory effects and triggers events related to tissue repair. However, it is essential to point out that, although the BDNF has less affinity to the p75 receptor, its activation results in a pro-inflammatory and degenerative effect that is crucial to effective regeneration and tissue repair.

### BDNF in Platelet-Rich Plasma (PRP)

After the injection of PRP in the injured site, endogenous thrombin or collagen fragments released into the intra-articular space activate platelets [91]. Once activated, various proteins released by platelets act as anti-catabolic and anti-inflammatory agents, such as IL-1RA which prevents IL-1β-mediated inflammation, while stimulating soluble TNF-RI receptors to prevent the effect of TNFα, and TGF-β which inhibits cartilage degradation [91,92].

Although it is clinically evident that PRP therapy has a potential role in augmenting tissue repair, studies on the mechanisms of action of the proteins released by PRP and their specific role in PRP treatment are still scarce. Indeed, most of the effects attributed to this therapy are based on physiological properties already evidenced by the various proteins present in platelet granules [91,93,94]. However, it is not yet clear whether the beneficial effects of PRP involve the participation of all proteins released during platelet activation or if a trophic factor mainly contributes to the effects induced by PRP [95]. It is plausible to hypothesize that proteins released by platelets contribute differently to the effects of PRP, and the effectiveness of therapy depends on a required amount of specific growth factors, which depend on the platelet’s concentration.

Despite the lack of studies on the mechanisms of action of PRP, platelets can store large amounts of TGF-β, serotonin, and VEGF [96,97]. Additionally, BDNF is present in platelets and is restricted to α-platelet granules released during tissue injury and inflammation. Its concentration is about 100 times greater than in neuronal cells [98,99].

Interestingly, BDNF concentrations in the synovial fluid of patients with osteoarthritis are significantly increased compared to healthy individuals [100], suggesting that the release of BDNF by joint tissue is a natural response to the repair of the injured joint tissue. This fact reinforces the concept that orthobiologic therapy stimulates the natural healing ability of tissue.

In this sense, the association of orthobiologic therapy, in particular PRP, with Pulsed Radiofrequency (PRF) could further reinforce joint tissue repair. However, no evidence in the literature supports this association with neuropathies or other peripheral nervous system injuries.

However, with these ideas in mind, we can hypothesize that the modulation in BDNF response provided by the action of PRF, may prevent more exacerbated inflammatory responses occasionally encountered in the use of orthobiologics, and may also provide a gain in the activation of reactive cascades that stimulate neuroplasticity and neural tissue regeneration, also helping to control extracellular glutamate levels, in addition to synergistic effects to the paracrine actions of some orthobiologics in controlling pain and modulating inflammation [30].

## 5. Radiofrequency Combined with Orthobiologics—Current Evidence

In fact, PRF has a better description and defined effect when used on neural structures, the effects in other tissues and joints seem to be promising; however, in the literature we need more high-quality studies. We have some trials, especially for knee osteoarthritis. In 2016, Yuan et al. evaluated 42 patients with knee OA treated with PRF versus corticosteroids. It was verified until 24 weeks that the group that used PRF significantly presented improvement in functional evaluation through WOMAC and pain. In addition, the markers of catabolism (MMP-3, IL-1, and TNF-α) show significantly decreased levels in the group of PRF in comparison to corticosteroids. These results showed superiority in relation to corticosteroid therapy and also the ability to alleviate clinical symptoms and change the catabolic environment in the knee joint [24].

In 2007 Sluijter et al., published an article using PRF for arthrogenic pain in refractory patients. In this case series, the authors described six clinical cases of the cervical facet joint pain, knee joint, sacroiliac joint, radiocarpal joint degeneration, shoulder, and atlantoaxial joint. In general, in the follow-up, all the patients reported good conditions, with mobility and no pain recurrence [101].

Fini et al., reported in a review evaluating some results of experimental and clinical studies suggesting that the use of electromagnetic field stimulation could be a promising chondroprotective therapy for OA joints. In these articles, it was related to an in vivo action on chondrocyte metabolism by enhancing cartilaginous and subchondral bone tissue properties. In addition, clinical studies showed the amelioration of clinical and radiographic observations with the use of pulsed electromagnetic fields [102].

According to the current knowledge, the combination of PRF and orthobiologic therapies can be applied to pain of nociceptive origin from damage to peripheral tissues. There is a lack of evidence about the combination and its possible effects. Michno et al. published an in vitro study using leukocyte-rich PRP (L-PRP) in combination with PRF, and it was observed that PRF was able to activate L-PRP in a similar way that thrombin does. Additionally, the PRF settings used (PRF energy at 500 kHz, at a pulse rate of 5 ms/5 Hz, for a period of 120 s, with an adjusted voltage of 40 V to permit a maximum temperature of 42 °C) had no impact on the functional integrity of platelets and leukocytes present in PRP, suggesting the safety of PRF but only for cellular components of PRP and in vitro conditions [103]. Aside from pain relief, some in vitro studies have demonstrated that low-frequency pulsed electromagnetic fields could enhance the healing of bone, cartilage, and tendon tissue [103,104,105]. Through these studies, it was possible to verify that external physical stimulations could enhance cells to produce extracellular matrix and cytokines for cell proliferation. Additionally, a superposition effect was observed when cells were co-cultured with PRP and electrical stimulus [106].

Clinical studies are case series publications with a small number of patients. In 2021, Jin et al., combined PRF with four PRP injections into the knee joint to treat OA. As result, a reduction in pain, and the recovery of knee joint mobility were observed. The purpose of the combined application of PRF and PRP would be to make full use of the analgesic effect of PRF technology in combination with the repairing effect of PRP, which can maximize the advantages of both treatment methods with minimally invasive techniques. In conclusion, it was verified that the combined application of PRF and PRP can promote the improvement of knee osteoarthritis symptoms and shorten the course of treatment which may restore knee joint function in the shortest time frame, which is an important outcome for patients [107].

The combination of PRF and PRP was also used to treat supraspinatus injury. A case report with four patients used PRP as a single application after the PRF procedure. As a result, the patients reported no significant increase in pain after the procedure. Additionally, the Constant–Murley shoulder score was increased and the range of movement was significantly improved, showing positive results with the combination of both procedures. All these outcomes were maintained for 6 months [108].

RF treatment, especially when ablative, uses heat to coagulate proteins, inactivate nerve endings, and eliminate edema, promoting a better analgesic effect. However, the repairing effect is not better than other conservative treatments. The efficacy of PRP treatment is based on the production of collagen and growth factors, increasing the number of local cells, and providing bioactive particles with powerful repair functions. Therefore, the use of both treatments may significantly augment the improvement of symptoms, and shorten the treatment course, with an improvement in life quality and a faster return to daily activities [108].

Filippiadis et al., published a study using PRF in combination with hyaluronic acid (HA) to treat knee osteoarthritis. Forty-five patients (53 knees) were included with a follow-up period of 12 months. The results showed an initial improvement in the first month in 86.8% of the knees. Pain recurrence during the follow-up period was noticed in 39.1% of the joints that, in the first month, responded well to treatment. Overall mobility was improved in 88.6% of the patients. The combination of PRF with HA seemed to further increase the duration of pain reduction [109].

In Filippiadis’ study, the use of intra-articular injections of HA aimed to provide a viscoelastic supplementation of the joint. As PRF does not promote osteophyte regression or cartilage and meniscus regeneration in patients with substantial bone and cartilage damage, the addition of hyaluronate seemed to be a necessity. In conclusion, the authors described that percutaneous, intra-articular application of PRF combined with HA is an effective and safe technique for the palliative management of chronic pain in patients with knee osteoarthritis. Results seem to be reproducible and long-lasting (1 year), which may require a repeat procedure after this period [109].

A single-blinded randomized clinical trial compared the use of intra-articular injection of PRP with pulsed radiofrequency of the genicular nerves in 200 patients with knee osteoarthritis. As a result, it was reported, that a significant difference in pain relief occurred between the groups as the PRF group showed a lower score than the PRP group [110].

The aesthetic field has also evaluated the combination of PRP with RF. A study published recently compared the effect of three sessions of fractional radiofrequency microneedling alone and in combination with PRP in 20 patients with mild to moderate neck laxity. The authors observed that both groups presented a statistically significant improvement in dermal thickness; however, the difference was not significant when the groups were compared [111]. A pilot study evaluated the efficacy of combining intradermal RF with PRP for the treatment of striae distensae. It was observed that the patients achieved excellent visual improvement and were satisfied with the treatment [112].

## 6. Conclusions

With this review, we hypothesized that the combination of PRF with orthobiologics may have potential benefits due to a faster improvement in pain, neuromodulation, and stimulus to extracellular matrix secretion (Figure 2). This combination aims to regenerate the injured tissue by stimulating endogenous repair through chemotaxis, delivery of growth factors, and the modulation of the lesion microenvironment. The potential effect and results may differ according to the type of orthobiologic used. However, based on the studies, we believe that an increased improvement in pain, function, and mobility, resulting in better life quality for the patients could be achieved.

## Figures and Tables

**Figure 1 ijms-23-11726-f001:**
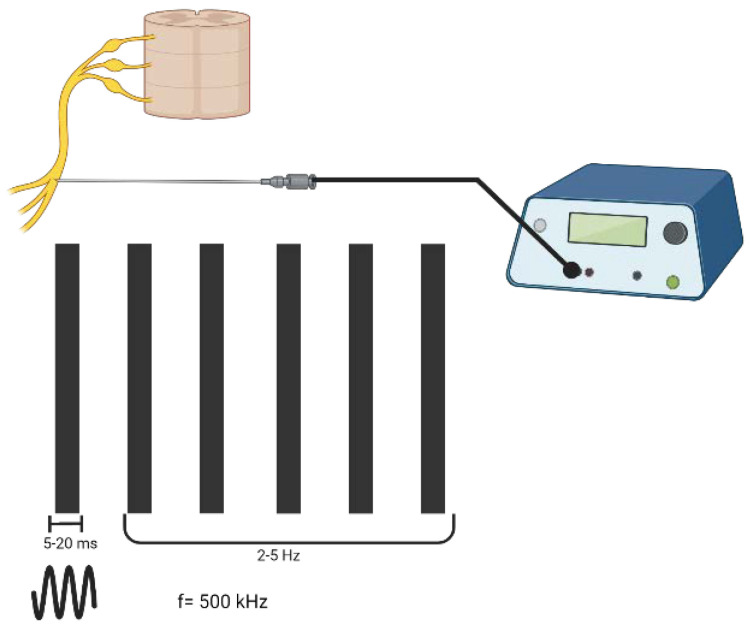
The parameters of pulsed radiofrequency; the way waveforms of electricity are delivered to tissues. Created with BioRender.com, accessed on 6 September 2022.

**Figure 2 ijms-23-11726-f002:**
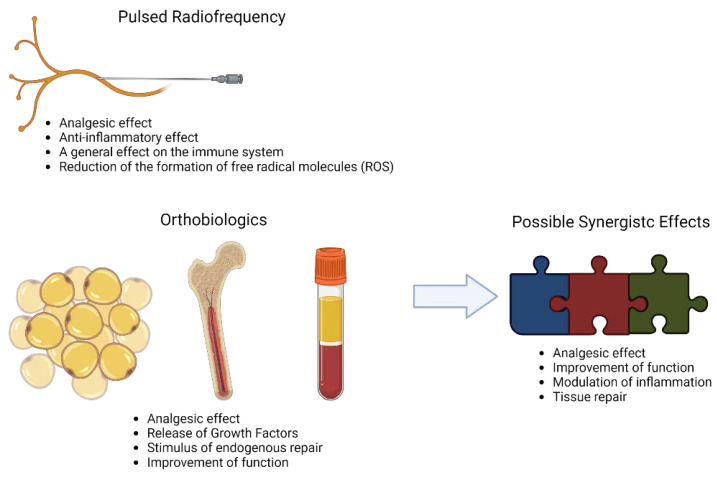
Synergistic proposal effects of PRF combined with orthobiologics. Created with BioRender.com, accessed on 6 September 2022.

**Table 1 ijms-23-11726-t001:** Analgesic effects of PRF related to nerves and other tissues and its mechanisms [30].

Nerves	Other Tissues
**Neurotransmitters** Glutamate and aspartate—inhibition by PRF—decrease nerve cell polarization GABA(B)-R1 and 5-HT3r upregulation of inhibitory pathways—reversed mechanical allodynia M-ENK upregulation—reduction of nociceptive stimulation	**Ions Channels** Dorsal root ganglion and spinal cord—analgesic effect by upregulating gene expression of Na/K channels Upregulation HCN—reduced hyperalgesia and allodynia Downregulation of P2X3 expression to alleviate nociceptive signals
	**Small Peptides** Modulation of CGRP
	**Inflammatory Cytokines** Modulation/decrease through different pathways of the cytokines: IL-6, IL-17, IFN-γ, IFR8, TNF-α, IGF-2
	**Intracellular proteins** Decreased levels of β-catenin Modulation of JNK

Abbreviations: GABA(B)-R1—Gamma-aminobutyric acid B receptor 1; (h-HT3r) 5-hydroxytryptamine receptor 3A; (M-ENK) metenkephalin; Na—sodium; K—potassium; (HCN) hyperpolarization-activated cyclic nucleotide-gated; (P2X3) Purinergic ligand-gated ion channel 3 (P2X3) receptor, (CGRP) Calcitonin-gene related product; IL-6 Interleukin 6; IL-17 Interleukin 17, IFN-γ Interferon gamma, IFR8 interferon regulatory factor 8, TNF-α, tumor necrosis factor, IGF-2 insulin growth factor 2; (JNK) c-Jun N-terminal kinases.

## Data Availability

Not applicable.

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
