# Peer review of "The Mechanism of Action between Pulsed Radiofrequency and Orthobiologics: Is There a Synergistic Effect?"

_ijms, 2022, doi:10.3390/ijms231911726_

Round 1
Reviewer 1 Report
Dear Author
This review is clinically significant as it identifies possible combinations of pulsed radiofrequency and orthobiologics.
It is carefully described throughout, but is more about other key words than the main topic.
Therefore, it is considered necessary to re-examine the following points.
・However, there are many descriptions outside of the main theme. Therefore, the sections need to be further summarised. Especially in this review "4. Brain-derived Neurotrophic Factor (BDNF) could be a critical molecule to deter-233 mine PRP-PRF-associated treatment: L233-323". Only the important points should be mentioned, as they seem to be of low importance.
Author Response
Dear Reviewers
International Journal of Molecular Sciences
September 07th 2022.
We have received your e-mail with the Reviewers’ comments regarding our manuscript, entitled “The mechanism of action between pulsed radiofrequency and orthobiologics: Is there a synergistic effect?” by Jorge and colleagues submitted to International Journal of Molecular Sciences. We have worked thoroughly to answer all queries raised by the Reviewers by point in order to improve the quality of our manuscript and make it suitable to be published at International Journal of Molecular Sciences. We are presenting below all the queries stated by the reviewer as well as the way we have dealt with them to perform the requested changes. We have prepared below a list that responds to the Reviewers’ comments.
This review is clinically significant as it identifies possible combinations of pulsed radiofrequency and orthobiologics.
It is carefully described throughout, but is more about other key words than the main topic.
Therefore, it is considered necessary to re-examine the following points.
・However, there are many descriptions outside of the main theme. Therefore, the sections need to be further summarised. Especially in this review "4. Brain-derived Neurotrophic Factor (BDNF) could be a critical molecule to deter-233 mine PRP-PRF-associated treatment: L233-323". Only the important points should be mentioned, as they seem to be of low importance.
Answer: We agree with this suggestion – we changed the topic, summarizing the main points regarding BDNF. The item 4 was changed from : Data from literature now consider Brain-derived Neurotrophic Factor (BDNF) a critical factor involved in the anti-hyperalgesic effect mediated by PRF applied in Dorsal Root Ganglion (DRG) or Spinal Cord. BDNF is a protein of the neurotrophin family. Neurotrophins are a family of growth factors that include, in addition to BDNF, nerve growth factor (NGF), neurotrophin-3 (NT-3), and neurotrophin 4/5 (NT-4/5). They were initially described as trophic factors involved in neuronal development, survival, and function [69, 70]. Developmentally, they are involved in synapse formation and neuroplasticity. This involvement was demonstrated in some chronic pain conditions of the nervous systems where the formation of a new synapse and neuroplasticity are maladaptive [71 – 74]. Each member of the neurotrophin family has a specific receptor for which the ligand (protein/neurotrophic factor) has a high affinity. BDNF exerts its physiological functions through two different types of receptors: Tyrosine kinase B (TrkB) with higher affinity and pan-neurotrophin receptor p75 (p75NTR) with lower affinity [75, 76]. The interaction of BDNF with the TrkB receptor provokes its dimerization, transphosphorylation of tyrosine residues, and subsequent activation of three main intracellular signaling pathways: 1) mitogen-activated kinases (MAPK) activation that act in cellular differentiation; 2) the phosphatidyl-inositol kinase (PI3K) activation promoting cellular survival and growth, and 3) the Phospholipase C-γ (PLC-γ) pathway activation [69, 70]. Outside the nervous system, neurotrophin receptors, including TrKB and P75, are expressed in tissues such as skeletal muscle, bone, immune system cells (basophils, eosinophils, lymphocytes; macrophages; mast cells, neutrophils), cartilage, and synovium [76 – 82]. In this wide variety of tissues, neurotrophins play a constitutive role in regulating cell homeostasis and tissue development [76, 79, 81]. In cartilage and synovium, although the role of neurotrophins is still not clear, there is an upregulation of BDNF in chondrocytes and synovial cells during an inflammatory process of the joint [81,83]. BDNF also stimulates the TrkB-Ras-ERK1/2-Elk-1 signaling pathway in bone tissue and regulates the expression of bone morphogenic protein-2 (BMP-2), resulting in the formation of new bone by osteoblasts [76]. Regarding the p75 receptor activation, some studies report a possible activity in the differentiation of osteoblasts from stem cells by stimulating the Rho-GTPase pathway [84,85]. The genetic deletion of p75 generates a permanent decrease in collagen formation, consequently inhibiting intramembranous and endochondral ossification [86]. In cartilage, BDNF regulates the activity of p38 and ERK-p44 kinases and increases the activity of transcription factors related to cell differentiation, such as SOX-9 and Runx-2, which also regulate cartilage matrix synthesis and tissue homeostasis. [78, 79, 87]. In synoviocytes, ATP, specifically via P2x4 receptors, regulates the expression of BDNF, increasing the expression of this protein during osteoarthritis conditions [88, 89]. Inflammatory conditions also increase p75 receptors in synoviocytes and chondrocytes [77, 90, 80]. However, it seems that p75 activation mediates different effects than TrKB receptor activation. While the activation of the p75 receptor in synovial cells, macrophages, and dendritic cells promotes a pro-inflammatory microenvironment, the TrKB receptor activation in macrophages and dendritic cells promotes an anti-inflammatory microenvironment. The P75 activation in synovial cells induces the release of MMP-2 metalloproteinase enzymes and the synthesis of pro-inflammatory cytokines, such as IL-1β, TNF-α, IL-6, and IL-8 [80]. Activation of the p75 receptor in macrophages promotes an increase of the synthesis of macrophage chemotactic protein (MCP-1), expression of MMP-9 metalloproteinase, and phagocytosis [91]. Regarding dendritic cells, p75 activation induces the synthesis of IFN-γ and IL-5 synthesis [92, 93]. On the other hand, the activation of TrKB receptor in macrophages induces the polarization of M2 macrophages and synthesis of anti-inflammatory cytokines IL-10 and IL-13 [94 – 97] and in dendritic cells, the synthesis and release of IL-4 [98]. In summary, data from the literature describes that BDNF outside the nervous system via TrKB receptor activation promotes anti-inflammatory and immunomodulatory effects and triggers the events related to tissue repair. However, it is essential to point out that, although the BDNF has less affinity to the p75 receptor, its activation resulting in a pro-inflammatory and degenerative effect is crucial to effective regeneration and tissue repair. To: “Data from literature now consider Brain-derived Neurotrophic Factor (BDNF) a critical factor involved in the anti-hyperalgesic effect mediated by PRF applied in Dorsal Root Ganglion (DRG) or Spinal Cord. BDNF is a protein of the neurotrophin family. Neurotrophins are a family of growth factors that include, in addition to BDNF, nerve growth factor (NGF), neurotrophin-3 (NT-3), and neurotrophin 4/5 (NT-4/5). They were initially described as trophic factors involved in neuronal development, survival, and function [69, 70]. Developmentally, they are involved in synapse formation and neuroplasticity. This involvement was demonstrated in some chronic pain conditions of the nervous systems where the formation of a new synapse and neuroplasticity are maladaptive [71 – 74]. BDNF exerts its physiological functions through two different types of receptors: Tyrosine kinase B (TrkB) with higher affinity and pan-neurotrophin receptor p75 (p75NTR) with lower affinity [75, 76]. Outside the nervous system, neurotrophin receptors, including TrKB and P75, are expressed in tissues such as skeletal muscle, bone, immune system cells (basophils, eosinophils, lymphocytes; macrophages; mast cells, neutrophils), cartilage, and synovium [76 – 82]. In this wide variety of tissues, neurotrophins play a constitutive role in regulating cell homeostasis and tissue development [76, 79, 81]. They exert different roles in every tissue, however, in general, TrKB receptors are related to cell differentiation, cell-matrix synthesis, and tissue homeostasis [78, 79, 87]. In addition, the activation of the TrKB receptor in macrophages induces the polarization of M2 macrophages and the synthesis of anti-inflammatory cytokines IL-10 and IL-13 [94 – 97], and in dendritic cells, the synthesis and release of IL-4 [98]. On the other hand, P75 receptors are related to a pro-inflammatory microenvironment with the release of MMP-2 metalloproteinase enzymes and the synthesis of pro-inflammatory cytokines, such as IL-1β, TNF-α, IL-6, and IL-8 [80], also, expression of MMP-9 metalloproteinase, and phagocytosis [91]. In summary, data from the literature describes that BDNF outside the nervous system via TrKB receptor activation promotes anti-inflammatory and immunomodulatory effects and triggers events related to tissue repair. However, it is essential to point out that, although the BDNF has less affinity to the p75 receptor, its activation resulting in a pro-inflammatory and degenerative effect is crucial to effective regeneration and tissue repair”.

Reviewer 2 Report
In this review article the authors explore the molecular effects of different analgesic treatments, namely pulsed radiofrequency and orthobiologics treatments, in order to suggest a possible synergic effect.
The idea is good, and the possible mechanism provided make sense and are well documented. However, concepts are put in a confusing way and the logical consequentiality is lacking. The effects on nociception, and inflammation must be clearly distinguished. Analgesia can derive from fading of inflammation and tissue repair, but analgesic effects of PRF may be independent from tissue repair, given the potential of reducing the nociceptive input. Here are some suggestions to improve the manuscript.
1. My first impression is that the application field that the authors suggest is somehow vague. For instance, PRF has an effect on nociception (i.e., it reduces transmission of small fibers and it can modulate synaptic transmission when applied directly to the nerve). However, its effect is of greater magnitude if the needle is placed on (or near) a ganglion. Less clear is the effect on tissues different from neural tissues, such as tendons and joints, I think this latter point needs to be explicated in the text. This concept is fundamental to explain in what pathology may the two techniques be synergic. I believe that the synergy can be at a peripheral level on tissue lesions due to mechanical issues (arthritis, joints overload, tendon shear or damage) with or without a clear inflammatory process. However, the two techniques can be used as an “add on therapy” in degenerative disease such as knee osteoarthritis, where PRP can be delivered and PRF can be used on nerves such as genicular nerves, or saphenous nerve (Carpenedo R et al, Pain Manag. 2022 Mar;12(2):181-193. doi: 10.2217/pmt-2021-0035) or DRG. As Cohen et al. aptly put it [Cohen SP, Van Zundert J. Editorial: Pulsed radiofrequency: Rebel without cause. Reg Anesth Pain Med. 2010;35(1):8-10. doi:10.1097/AAP.0b013e3181c7705f], “PRF is a treatment in search for a cause”. Researchers must establish the scientific rationale behind the use of PRF in all patients with pain of different etiologies. In sum, I would add a sentence in the “current evidence” paragraph that states that PRF has a better understood and defined effect when applied on neural structures and that its effects on tissues and joints seems to be promising, but it has to be better explored. However, given the current knowledge, the combination of PRF and orthobiologic therapies can be applied on pain of nociceptive origin from damage of peripheral tissues. Here you can also make some suggestion of possible roles in the future.
2. Line 42-43: “The application of PRF is based on the delivery of a train of sinusoidal electrical bursts (5-20 ms length) in the radiofrequency range (500 kHz) at a repetitive rate of a few hertz (2-5 Hz).” A picture will help the reader to understand in which waveforms electricity is delivered to tissues
3. A table will help the reader to understand the possible analgesic effects of PRF. I suggest dividing items by effects on nerves and effects on other tissues. A picture may also help
4. Figure 1 is too simple. Try to summarize synergic effects: tissue repair, inflammation fading, reduction of nociceptive inputs, reduction of pain sensitization… Divide analgesic effects from effects on inflammation/repairing systems.
Author Response
In this review article the authors explore the molecular effects of different analgesic treatments, namely pulsed radiofrequency and orthobiologics treatments, in order to suggest a possible synergic effect.
The idea is good, and the possible mechanism provided make sense and are well documented. However, concepts are put in a confusing way and the logical consequentiality is lacking. The effects on nociception, and inflammation must be clearly distinguished. Analgesia can derive from fading of inflammation and tissue repair, but analgesic effects of PRF may be independent from tissue repair, given the potential of reducing the nociceptive input. Here are some suggestions to improve the manuscript.
- My first impression is that the application field that the authors suggest is somehow vague. For instance, PRF has an effect on nociception (i.e., it reduces transmission of small fibers and it can modulate synaptic transmission when applied directly to the nerve). However, its effect is of greater magnitude if the needle is placed on (or near) a ganglion. Less clear is the effect on tissues different from neural tissues, such as tendons and joints, I think this latter point needs to be explicated in the text. This concept is fundamental to explain in what pathology may the two techniques be synergic. I believe that the synergy can be at a peripheral level on tissue lesions due to mechanical issues (arthritis, joints overload, tendon shear or damage) with or without a clear inflammatory process. However, the two techniques can be used as an “add on therapy” in degenerative disease such as knee osteoarthritis, where PRP can be delivered and PRF can be used on nerves such as genicular nerves, or saphenous nerve (Carpenedo R et al, Pain Manag. 2022 Mar;12(2):181-193. doi: 10.2217/pmt-2021-0035) or DRG. As Cohen et al. aptly put it [Cohen SP, Van Zundert J. Editorial: Pulsed radiofrequency: Rebel without cause. Reg Anesth Pain Med. 2010;35(1):8-10. doi:10.1097/AAP.0b013e3181c7705f], “PRF is a treatment in search for a cause”. Researchers must establish the scientific rationale behind the use of PRF in all patients with pain of different etiologies. In sum, I would add a sentence in the “current evidence” paragraph that states that PRF has a better understood and defined effect when applied on neural structures and that its effects on tissues and joints seems to be promising, but it has to be better explored. However, given the current knowledge, the combination of PRF and orthobiologic therapies can be applied on pain of nociceptive origin from damage to peripheral tissues. Here you can also make some suggestion of possible roles in the future.
Answer: We agree with your suggestion. We change the topic 5. Radiofrequency Combined with Orthobiologics – Current Evidence. At the beginning, line 304 – it was added some paragraphs talking about the current evidence of PRF in joints. Line 304: “In fact, PRF has a better description and defined effect when used on neural structures, the effects in other tissues and joints seem to be promising, however, in the literature we need more high-quality studies. We have some trials, especially for knee osteoarthritis. In 2016, Yuan et al, evaluated 42 patients with knee OA treated with PRF versus corticosteroids. It was verified until 24 weeks that the group that used PRF significantly presented improvement in functional evaluation through WOMAC and pain. In addition, the markers of catabolism (MMP-3, IL-1 and TNF-α) show significantly decreased levels in the group of PRF in comparison to corticosteroids. These results showed superiority in relation to corticosteroid therapy and also the ability to alleviate clinical symptoms and change the catabolic environment in the knee joint. [Yuan, Y. et al., 2016] . In 2007 Sluijter et al., published an article using PRF for arthrogenic pain in refractory patients. In this case series, the authors described 6 clinical cases of cervical facet joint pain, knee joint, sacroiliac joint, radiocarpal joint degeneration, shoulder, atlanto-axial joint. In general, in the follow-up all the patients reported good conditions, with mobility and no pain recurrence [Sluijter ME, et al., 2008].
Fini et al., 2005 reported in a review, evaluating some results of experimental and clinical studies suggesting that the use of electromagnetic field stimulation could be a promising chondroprotective therapy for OA joints. In these articles, it was related to an in vivo action on chondrocyte metabolism by enhancing cartilaginous and subchondral bone tissue properties. In addition, clinical studies showed the amelioration of clinical and radiographic observations with the use of pulsed electromagnetic fields. [ Fini et al., 2005].
According to the current knowledge, the combination of PRF and orthobiologic therapies can be applied to pain of nociceptive origin from damage to peripheral tissues. There is some evidence about the combination and its possible effects”. Cited articles with the combination described in this section 5.
- Line 42-43: “The application of PRF is based on the delivery of a train of sinusoidal electrical bursts (5-20 ms length) in the radiofrequency range (500 kHz) at a repetitive rate of a few hertz (2-5 Hz).” A picture will help the reader to understand in which waveforms electricity is delivered to tissues.
Answer: We agree with the reviewer’s suggestion. So, it was added figure one, with the illustration of pulsed radiofrequency parameters.
- A tablewill help the reader to understand the possible analgesic effects of PRF. I suggest dividing items by effects on nerves and effects on other tissues. A picture may also help
Answer: We agree with the reviewer’s suggestion. It was added a table (Table 1) with the molecules and changes related to pain in nerves and other tissues.
|
Nerves |
Other tissues |
|
Neurotransmitters Glutamate and aspartate – inhibition by PRF – decrease nerve cell polarization GABA(B)-R1 and 5-HT3r upregulation of inhibitory pathways - reversed mechanical allodynia M-ENK upregulation – reduction of nociceptive stimulation |
Ions Channels Dorsal root ganglion and spinal cord – analgesic effect by upregulating gene expression of Na/K channels Upregulation HCN – reduced hyperalgesia and allodynia Downregulation of P2X3 expression to alleviate nociceptive signals |
|
|
Small Peptides Modulation of CGRP |
|
|
Inflammatory Cytokines Modulation/decrease through different pathways of the cytokines: IL-6, IL-17, IFN- γ, IFR8, TNF-α, IGF-2 |
|
|
Intracellular proteins Decreased levels of β-catenin Modulation of JNK |
Abbreviations: GABA(B)-R1 - Gamma-aminobutyric acid B receptor 1; (h-HT3r) 5-hydroxytryptamine receptor 3ª; (M-ENK) metenkephalin; Na – sodium; K – potassium; (HCN) hyperpolarization-activated cyclic nucleotide-gated; ( P2X3) Purinergic ligand-gated ion channel 3 (P2X3) receptor, (CGRP) Calcitonin-gene related product; IL-6 Interleukin 6; IL-17 Interleukin 17, IFN- γ Interferon gamma, IFR8 interferon regulatory factor 8, TNF-α, tumor necrosis factor, IGF-2 insulin growth factor 2; (JNK) c-Jun N-terminal kinases [30].
- Figure 1 is too simple. Try to summarize synergic effects: tissue repair, inflammation fading, reduction of nociceptive inputs, reduction of pain sensitization… Divide analgesic effects from effects on inflammation/repairing systems.
Answer: The figure 1 (now 2) was changed to better describe the possible synergistic effects with the combination of orthobiologics and PRF.
Thank you very much for your consideration; I look forward to hearing from you in due course.
Sincerely,
Daniel Jorge, MD
